# End-To-End Deep Learning Architecture for Continuous Blood Pressure Estimation Using Attention Mechanism

**DOI:** 10.3390/s20082338

**Published:** 2020-04-20

**Authors:** Heesang Eom, Dongseok Lee, Seungwoo Han, Yuli Sun Hariyani, Yonggyu Lim, Illsoo Sohn, Kwangsuk Park, Cheolsoo Park

**Affiliations:** 1Department of Computer Engineering, Kwangwoon University, Seoul 01897, Korea; surmounting@kw.ac.kr (H.E.); yulisun@telkomuniversity.ac.id (Y.S.H.); 2Interdisciplinary Program in Bioengineering, Seoul National University, Seoul 03080, Korea; azuremoon@bmsil.snu.ac.kr; 3Department of Intelligent Information System and Embedded Software Engineering, Kwangwoon University, Seoul 01897, Korea; seungwoohan@kw.ac.kr; 4School of Applied Science, Telkom University, Bandung 40257, Indonesia; 5Department of Oriental Biomedical Engineering, Sangji University, Wonju 26339, Korea; yglim@sangji.ac.kr; 6Department of Computer Science and Engineering, Seoul National University of Science and Technology, Seoul 01811, Korea; isohn@seoultech.ac.kr; 7Department of Biomedical Engineering, College of Medicine, Seoul National University, Seoul 03080, Korea; 8Institute of Medical and Biological Engineering, Medical Research Center, Seoul National University, Seoul 03080, Korea

**Keywords:** blood pressure, electrocardiogram, photoplethysmogram, ballistocardiogram, deep learning, signal processing, attention mechanism

## Abstract

Blood pressure (BP) is a vital sign that provides fundamental health information regarding patients. Continuous BP monitoring is important for patients with hypertension. Various studies have proposed cuff-less BP monitoring methods using pulse transit time. We propose an end-to-end deep learning architecture using only raw signals without the process of extracting features to improve the BP estimation performance using the attention mechanism. The proposed model consisted of a convolutional neural network, a bidirectional gated recurrent unit, and an attention mechanism. The model was trained by a calibration-based method, using the data of each subject. The performance of the model was compared to the model that used each combination of the three signals, and the model with the attention mechanism showed better performance than other state-of-the-art methods, including conventional linear regression method using pulse transit time (PTT). A total of 15 subjects were recruited, and electrocardiogram, ballistocardiogram, and photoplethysmogram levels were measured. The 95% confidence interval of the reference BP was [86.34, 143.74] and [51.28, 88.74] for systolic BP (SBP) and diastolic BP (DBP), respectively. The $R^{2}$ values were 0.52 and 0.49, and the mean-absolute-error values were 4.06 ± 4.04 and 3.33 ± 3.42 for SBP and DBP, respectively. In addition, the results complied with global standards. The results show the applicability of the proposed model as an analytical metric for BP estimation.

## 1. Introduction

Blood pressure (BP) is one of the vital signs that provide fundamental health information of the patient. When the heart beats, the blood flow from the heart exerts a pressure on the blood vessels. BP varies between systolic BP (SBP), which is an increase in pressure caused by systolic contraction of the heart, and diastolic BP (DBP), which is the low pressure between SBPs. High BP (hypertension) is known as a “silent killer” because it is a risk factor for various diseases such as arrhythmia, heart attack, blindness, and brain stroke. An estimated 1.13 billion people worldwide have hypertension [1].

The gold standard of BP monitoring is a sphygmomanometer, which is usually used in the physician’s office. However, BP measurement in medical circumstances may be incorrect because, in a clinical setting, some patients exhibit a higher BP than usual, which is known as the “white-coat effect.” The arterial BP (ABP) is considered as a gold standard for continuous BP monitoring, which is performed in the intensive care unit. However, measurement of ABP can cause side effects such as bleeding and infection because it is an invasive method that requires the insertion of an intravascular cannula needle. Moreover, it is difficult to measure ABP in daily life because it requires a clinical setting.

Because regular BP monitoring is important for the diagnosis of hypertension and the prediction of heart diseases, numerous devices have been created to help patients measure BP at home or during their daily lives. These devices are usually based on the oscillometric method and use an inflatable upper-arm cuff [2,3]. However, these methods do not offer continuous measurement. Currently, the significance of beat-to-beat BP analysis has increased, such as blood pressure variability (BPV) [4]. In addition, wearing the cuff whenever the patient tries to measure BP is a cumbersome task and can make the patient feel uncomfortable.

To overcome these limitations, several researchers have investigated cuff-less and continuous BP monitoring methods using physiological signals. The BP can be estimated based on the pulse wave velocity (PWV), which is the velocity of the arterial blood wave [5]. PWV can be calculated from the pulse transit time (PTT) as follows:(1)PWV=L/PTT
where L denotes the distance between two places where the blood wave propagates and PTT is the time that the blood wave takes to travel between two places.

The most frequently used definition of PTT is the time interval between the R peak of the electrocardiogram (ECG) and the systolic peak of the photoplethysmogram (PPG), which is measured at the finger. PTT from ECG and PPG can be easily applied in long-term monitoring or daily life situations because both ECG and PPG are measured noninvasively and do not require a medical expert or a cuff. PTT is known to be negatively correlated with BP [6], and various models have been developed to estimate BP using PTT [5,7,8,9,10].

In addition, some studies have suggested ballistocardiograms (BCGs) to estimate BP [11]. A BCG is a measurement of the forces of the body to the blood flow ejected from the heart after each heartbeat. BCG signals can be acquired using force sensors such as an accelerometer, load-cell, and polyvinylidene fluoride (PVDF) film sensor. This method enables unobtrusive BP estimation because BCG is a noninvasive and unconstrained method to monitor cardiovascular activity. Shin et al. suggested a system for BP estimation using a weighing scale [12]. ECG and BCG were measured on a weighing scale, and the R-J interval (RJI) was calculated between the R peak of the ECG and the J peak of the BCG signal. The results showed that RJI was negatively correlated with BP. Lee et al. proposed a chair-based BP monitoring system using a two channel BCG [13]. BP was estimated using the phase difference between two BCGs measured at the back of the chair and the cushion on the seat.

Other studies have proposed using machine learning (including deep learning) algorithms to estimate continuous BP without a cuff automatically. Chan et al. [14] and Kachuee et al. [15] proposed a model for estimating BP based on features extracted from ECG and PPG using conventional machine learning algorithms such as linear regression and AdaBoost. Su et al. proposed an recurrent neural network (RNN) based BP estimation model using features extracted from ECG and PPG [16]. Kurylyak et al. [17], Lee et al. [13], and Wang et al. [18] also proposed a simple artificial neural network (ANN) model that used features extracted from only a single signal, such as PPG or BCG. However, these methods still have problems that the extraction of features is costly and a laborious task. In addition, if the signal is noisy, it may be difficult to obtain enough data to train the neural network.

In recent studies, some authors have attempted to estimate BP using raw signals without feature engineering. Slapnivcar et al. estimated BP with only raw PPG using ResNet, a deep learning model that performed well in the field of image classification [19] but had lower performance compared to the models used in other studies. Tanveer et al. also used ECG and PPG raw signals and achieved good performance [20]. However, the length of the data used for BP estimation was very long, 16 s, and, accordingly, the number of samples used for the overall estimation was small, making it difficult to evaluate performance accurately.

In this study, we developed an end-to-end deep learning model for BP estimation without feature extraction and used three physiological signals (ECG, PPG, and BCG) that were used in previous studies for estimating BP. The advantages of this study are as follows:BP can be estimated using only raw signals with minimal preprocessing.All combinations of signals were used as input, and their performance studied.By using the attention mechanism, the performance of the model was improved and its applicability as an analytical metric for BP estimation verified.

## 2. Materials and Methods

The overall experiment process is shown in Figure 1. The details of each step are explained in the following sections.

### 2.1. Data Acquisition

A total of 15 subjects (6 men, 9 women, age: 26.2±3.0) were recruited for this study. No medical records of the subjects were reported. Written informed consent was obtained from the subjects, and this study was approved by the Institutional Review Board of Seoul National University Hospital (IRB No. 1801-016-912).

The experimental setup is illustrated in Figure 2. Three Ag/AgCl electrodes were attached to the subject’s left arm, right arm, and left leg, according to Einthoven’s triangle. ECG was acquired on lead II using the electrodes with the BIOPAC ECG100C module, and PPG was measured at the subject’s index finger using a commercial module (PSL-iPPG2C) [21]. In addition, the BCG signals were measured using a PVDF film sensor attached on the seat of the chair. SBP and DBP were also measured simultaneously using a continuous BP monitoring device (Finometer Pro (Finapres Medical Systems, Enschede, The Netherlands)). All the data were synchronized and sampled at 1000 Hz with a data acquisition device (BIOPAC MP150 module (BIOPAC Systems Inc., Goleta, CA, USA)).

After the measurement device was attached, the subjects were asked to sit in the armchair. The data were measured after 30 min, while the subjects rested. The measured BP values are shown in Figure 3. The mean and standard deviation (SD) of SBP and DBP were 115.04±14.64 mmHg and 70.01±9.56 mmHg, respectively.

### 2.2. Data Preprocessing

A second-order Butterworth bandpass filter was applied to the data to remove baseline wandering and power-line noise. The cutoff frequency of each filter applied to each signal is summarized in Table 1.

All combinations of the raw signals (ECG, PPG, and BCG) were used as input for the deep learning model to investigate the effect of each signal on BP estimation. To ensure sufficient information was provided as input, 5 s segments of the signals were used as input to the model with an overlap of 1 ms. Since a large amount of data were used as input, the data were resampled to 125 Hz for efficient learning, which is equivalent to the sampling rate of the Physionet MIMIC database containing ECG and PPG signals [22]. The sampling rate of the BCG signal was also resampled to 125 Hz with reference to [23]. The target labels were set as BP values (SBP, DBP) corresponding to the end of each segment, as shown in Figure 4. The BP values that were not in the range between the mean ± 1.96 SD were considered as outliers and were eliminated.

### 2.3. Deep Learning Model

#### 2.3.1. Convolutional Neural Network

Convolutional neural networks (CNNs) have achieved considerable success in various challenging areas by extracting key features from a large amount of data such as images. Recently, CNNs have achieved remarkable performance in signal processing, especially in biomedical areas [24,25,26,27,28]. In this study, a CNN was used to extract key patterns automatically from the input signal.

#### 2.3.2. Bidirectional Gated Recurrent Unit

RNNs have been applied for sequential modeling in natural language understanding and video processing. However, conventional RNNs have the problem of the gradient exploding or vanishing when the sequence is long [29]. To solve this long-term dependency problem, two representative RNN-based models have been proposed. One is the long short term memory (LSTM) developed by Hochreiter and Schmidhuber [30], and the other is the gated recurrent unit (GRU) introduced by Cho et al. [31]. Although both models have achieved much better performance than conventional RNN [32], the LSTM-based model requires more training time as it has more learning parameters than GRU. Therefore, the GRU layer was used in our model. The internal structure of GRU used in this study is illustrated in Figure 5.

There are two input vectors in a GRU cell at each time step, which include the previous hidden output vector $h_{t-1}$ and the current input vector $x_{t}$. The equation of the current hidden output vector $h_{t}$ can be expressed as follows:(2)$$\begin{array}{c} z_{t}=\sigma \left(W_{z}\cdot \left[h_{t-1},x_{t}\right]\right)\\ r_{t}=\sigma \left(W_{r}\cdot \left[h_{t-1},x_{t}\right]\right)\\ {\tilde{h}}_{t}=\tanh\left(W_{h}\cdot \left[r_{t}⊗h_{t-1},x_{t}\right]\right)\\ h_{t}=\left(1-z_{t}\right)⊗h_{t-1}+z_{t}⊗{\tilde{h}}_{t} \end{array}$$
where $r_{t}$ and $z_{t}$ are the update gate and the reset gate vector, respectively. $W_{z}$, $W_{r}$, and $W_{h}$ are trainable weight parameters of each gate. ${\tilde{h}}_{t}$ is the candidate state that determines how much present information needs to be learned after the reset gate. σ(·) is a sigmoid function, and tanh(·) is a hyperbolic tangential function. The symbol ⊗ denotes element-wise multiplication.

Conventional GRU is unidirectional, which means that each hidden state in a GRU cell only considers past information. Unlike unidirectional GRU, a bidirectional GRU (Bi-GRU) uses the information from both future and past feature vectors. Bi-GRU was used in this study instead of the conventional GRU. The 5 s segment was used as input in our model; hence, meaningful information existed in both directions at a specific time.

The Bi-GRU has two layers, forward and backward layers, as shown in Figure 6. The final output hidden vector $h_{t}$ can be represented as a concatenation of the forward hidden vector $\vec{h_{t}}$ and backward hidden vector $\overset{\leftarrow }{h_{t}}$, as shown:(3)$$h_{t}=\vec{h_{t}}⊕\overset{\leftarrow }{h_{t}}$$

#### 2.3.3. Attention Mechanism

The main principle of the attention mechanism is that the model learns by focusing on the region of interest. The attention mechanism has recently been shown to be efficient in image captioning [33], neural machine translation [34,35], and signal processing [36]. Because sequential feature vectors from Bi-GRU in the proposed model may contribute differently for estimating BP values, we added an attention model to automatically train how much the feature vectors were important in each time step. Larger weights can be assigned to significant information using the attention mechanism, and various methods to apply the attention mechanism have been proposed [37]. In this work, the feed-forward attention model [38] was used, which is also known as the self-attention model. Given the Bi-GRU hidden state vector $h_{i}$ at every time step i∈[1,N], the importance score $s_{i}$ was calculated through a score function score(·) as follows:(4)$$s_{i}=score\left(W_{s}h_{i}\;+\;b\right)$$
where Equation (Equation 4) can be represented as a single-layer perceptron having a trainable weight $W_{s}$ and a bias *b*. The score function can be set as an activation function in the neural network, and the tanh function was used in the proposed model.

After obtaining each importance score $s_{i}$ for the hidden state vector $h_{i}$, the attention weight $a_{i}$ was evaluated using softmax function, expressed as:(5)$$a_{i}=softmax\left(s_{i}\right)=\frac{\exp(s_{i})}{\sum_{i}^{}\exp\left(s_{i}\right)}$$
The final output vector v was obtained by calculating the weighted sum of the attention weight vector and the corresponding hidden state vector, as follows:(6)$$v=\sum_{i}^{N}a_{i}h_{i}$$

### 2.4. Proposed Model

#### 2.4.1. Model Architecture

Our proposed end-to-end deep learning network consists of a CNN layer, a Bi-GRU layer, and an attention layer, as shown in Figure 7. The deep learning models (including CNN and RNN) have shown promising performance in image-based time series recognition [39] and biosignal processing [25,40,41,42]. The attention mechanism has also been successfully applied to natural language processing [43].

In our proposed model, the CNN structure shown in Figure 8 was designed by referring to the VGGNet structure [44]. A total of 10 convolution layers with a rectified linear unit (ReLU) activation function were used to extract spatial pattern vectors from signals. Each convolution layer was followed by a batch normalization layer to reduce the internal covariate shift [45]. The last layer of each convolution module was set to a max-pooling layer to reduce the length of the inputs. The same padding was applied to each convolution operation. Both kernel and pooling size were set to 3, and the output channel size of the convolution layer was scaled up by a factor of 2 from 64 to 512 as it passed through each convolution module.

In the Bi-GRU phase, 64 hidden nodes were set up in each of the forward and backward layers, and 128 features were generated at each time step. The Bi-GRU layer can encode temporal information between features in our proposed model. In addition, representative features can be acquired in the Bi-GRU layer by reducing feature dimensions from 512 to 128.

The output of the Bi-GRU hidden state vector is weighted by Equations (Equation 4)–(Equation 6), and, finally, these vectors are summed, and then the SBP and DBP values come out through 1-layer perceptron. The detailed structure of the proposed model is shown in Table 2.

#### 2.4.2. Training Setting

Seventy percent of the dataset was used for training, 10% for validation, and 20% for testing. The Adam optimizer [46] was used with a learning rate of $10^{-3}$ and decay of $10^{-4}$ to optimize the hyperparameters of the model. The learning rate was set to the optimal value empirically, and the initial weights were selected randomly. The mean squared error (MSE) was used for the loss function. The model was trained with the early stopping method with the patience of 10 in a maximum of 50 epochs, and the batch size was set to 512. As a computing environment for network training, the Keras deep learning framework with TensorFlow backend and NVIDIA GeForce RTX 2080Ti (NVIDIA corporation, Santa Clara, CA, USA) (with 11 GB VRAM) was used.

## 3. Results

In this section, three types of results are presented. First, the BP estimation results are compared for each combination of signals. Second, the performances of the models with and without attention are compared when all signals were used as inputs. Finally, the performance of our end-to-end deep learning model was compared with that of the multiple linear regression (MLR) model that used interval features from the characteristic point of each signal. The root-mean-square error (RMSE) and mean absolute error (MAE) were used as metrics of BP estimation accuracy. In addition, coefficient of determination ($R^{2}$) values were calculated between the reference and estimated BP for all results. In addition, the Bland–Altman plot [47] was presented to increase the reliability of the results.

### 3.1. Performance Comparison by Signal Combination

A summary of the results from all combinations of the three signals is shown in Table 3. The model with ECG+PPG and ECG+PPG+BCG showed better performance for estimating BP than the other models.

When a combination of multiple signals (ECG+PPG, ECG+BCG, PPG+BCG, and ECG+PPG+BCG) was used as input, the estimation accuracy was much better than the cases when a single signal (ECG, PPG, and BCG) was used as input. The detailed values are shown in Table 4.

A repeated measurement analysis of variance (ANOVA) test was used to compare the performance when the combination of inputs was varied, and the differences between models were significant in both SBP and DBP estimation (*p* < 0.01). In addition, a paired *t*-test was performed between the results to compare each models. As shown in Table 5 and Table 6, the proposed model statistically outperformed the other methods regardless of the input in both SBP and DBP estimation.

### 3.2. Attention Mechanism Performance

As shown in Table 3, the model with the attention mechanism showed lower RMSE and MAE values than the model without the attention mechanism. An example of BP estimation results is presented in Figure 9. The model with the attention mechanism estimated the fluctuation in BP more precisely than the model without the attention mechanism. The accuracy of the model can be improved with the attention mechanism by learning more intensively in the time step that contains relatively important information in the 5 s input data.

In addition, we found that the attention weight was high at a specific time by investigating the attention heat map. The process of generating the attention heat map is as follows: The length of the data was reduced from 625 (length resampled to 125 Hz) to 8 by the CNN pooling layer, and thus Bi-GRU generated eight hidden state vectors. Accordingly, the original signal could be divided into eight sections, and the attention heat map was generated by assigning the attention weight obtained from the attention layer to each section.

The ANOVA test was conducted to interpret the attention weights of each timestep, and the results are summarized in Figure 10. The differences between timestep were statistically significant (*p* < 0.001), and the weights from timestep 2 to timestep 6 were significantly higher than other timesteps. The timestep 1 to timestep 7 includes 648 ms of feature information, and timestep 8 has 464 ms of feature information. Therefore, it means that the data from timestep 2 to timestep 6 (from 1.112 s to 3.704 s before the target BP) had more meaningful information to estimated BP.

Figure 11 shows the Bland–Altman plot between the estimated and reference BP. The limits of agreement (LOA) at 95% confidence intervals for DBP and SBP were measured as [–9.50, 9.50] and [–11.24, 11.63], respectively. This means that 95% of the error was within [lower LOA, upper LOA]. Moreover, the mean error values between estimation and reference were 0.03 and 0.20 for DBP and SBP estimation, respectively, which means that the model had very little bias.

Figure 12 shows the BP estimation result from two subjects. Overall, the estimated BP was similar to the reference BP, as shown in Figure 12a. However, as shown in Figure 12b, the error was high in cases where BP rapidly changes in a short time or if the overall range of BP is wide.

### 3.3. Comparison to the Multiple Linear Regression Model

The results of the proposed model were compared to the MLR model. The characteristic points from ECG, BCG, and PPG were detected to extract the features for the MLR model. ECG R-peak and the peak of the 1st derivative PPG were detected based on Pan and Tompkins’s algorithm [48], and the BCG J peak was detected by finding the highest peak between 110 ms to 250 ms after each R-peak. False-positive peaks were excluded manually. Then, the R-R interval (RRI), PTT, and RJI were calculated from each cardiac cycle (Figure 13a). In addition, 21 ± 11% of the cardiac cycles were excluded in cases where the peak was not detected because of motion artifacts (Figure 13b). The features were utilized as inputs of the MLR model, and SBP and DBP were used as dependent variables of the MLR model.

As summarized in Table 7, the MLR model showed lower performance for BP estimation than the proposed model. The input parameters of the MLR model were RRI, RJI, and PTT from ECG, PPG, and BCG, respectively. Although the input signal was the same in the MLR and proposed model, the MAE value of the MLR model was statistically higher in both SBP and DBP estimation (*p* < 0.05).

The scatter plots between PTT and SBP of the two cases are presented in Figure 14. Although the CC values between PTT and SBP were high in both cases, the $R^{2}$ value was 0.29 in the good case and only 0.05 in the bad case. The BP variation was not explained by the interval features alone, such as PTT, but our model could extract other features as well as the interval feature.

## 4. Discussion

### 4.1. Main Contributions

In this study, we proposed an end-to-end deep learning architecture that can estimate BP continuously using only raw signals without using feature engineering, which is time consuming and labor-intensive. In addition, we measured all the physiological signals such as ECG, PPG, and BCG mainly used in the previous BP estimation studies and compared the performances using all combinations of the signals as inputs to the model. Through these experiments, it was demonstrated that multiple signal involvement could improve the accuracy of the BP prediction over a single signal analysis. The proposed model achieved better performance than the MLR method using the conventional features, such as PTT and RJI. While the MLR model used features within only a bit corresponding to the BP to be estimated, the proposed model utilized multiple bits in 5 s length and learned the temporal relationship among the bits. Finally, we added the attention mechanism to improve the performance of the model. The results show that the data from previous bits as well as the bit right before the BP contributed to the estimation of the BP.

### 4.2. Result Interpretation from Global Standard Perspective of BP Monitoring

There are two types of BP monitoring global standards used as a performance indicators in most studies. One is the Association for the Advancement of the Medical Instrumentation (AAMI) standard [49], and the other is the British Hypertension Society (BHS) standard [50]. Table 8 and Table 9 show the results of using the proposed model against these standards. Though the study population of this study was 15, which was insufficient because the AAMI and BHS standards require at least 85 subjects, the results complied with the AAMI and BHS standards in grade A for both SBP and DBP.

### 4.3. Comparison Result With Related Works

The performances of the models used in related studies are shown in Table 10. Chan et al. proposed a linear regression model based on PTT [14]. Even though the study used mean error, the error was high and the data size was unspecified. Kachuee et al. [15], Kurylyak et al. [17], Su et al. [16], and Wang et al. [18] also proposed feature-based approaches. The studies have suggested various methods to extract the features, which need a lot of time. On the other hand, our proposed model used only a bandpass filtering and did not require a complicated feature extraction process.

Lee et al. proposed a BP estimation model using only two channels of BCG [13]. Though estimating BP using only BCG was meaningful, the evaluation was conducted with the one time BP measurement, not a continuous BP measurement. Tanveer et al. achieved excellent performance using raw signals from ECG and PPG [20], but the window length to estimate BP was very long, 16 s, while we used 5 s input. In addition, the authors noted that the performance of their proposed model is highly dependent on the proper division of ECG and PPG cycles. It means that additional work such as peak detection is required, and the quality of the signal affects the performance of model. Slapnivcar et al. also proposed a deep learning model using the PPG raw signal [19], but its performance was low compared to other studies. In addition, it is difficult to evaluate the model accurately because other evaluation metrics were not presented.

### 4.4. Limitations of the Study

Kauchee et al. [15] and Slapnivcar et al. [19] proposed a calibration-free (cal-free) model that did not require a calibration of each subject. In this study, we focused on a calibration-based (cal-based) model, but designing a cal-free model that is not affected by subject characteristics is another important issue. Thus, we also attempted the development of a cal-free method using leave-one-subject-out (LOSO) method, and the results were summarized in Table 11. The performance of the cal-free model was inferior to that of a cal-based model. In addition, the result did not comply with the global standard mentioned in Section 4.2 and needs to be improved in further study.

In this study, data were measured in a short time of 30 min, hence the performance of the method while predicting long-term BP should be considered. In addition, as mentioned in Section 4.2, the number of subjects (global standard: minimum of 85 subjects) was not sufficient; however, we used inputs with 5 s lengths and total sample size was 26,600,014, which was enough to evaluate the model. In addition, the data of patients with hypertension were not included in the study because most of the subjects were healthy 20-year-olds with no diseases. However, the 95% confidence interval of the BP that we used in the experiment was [86.34, 143.74] and [51.28, 88.74] for SBP and DBP, respectively. Moreover, about 9% and 5% of the BP data were in hypertension stage 1 and stage 2 ranges [51], though no subject was diagnosed as a hypertension patient.

## 5. Conclusions and Future Work

In this paper, we proposed an end-to-end deep learning algorithm to estimate BP with attention mechanism. The proposed method was designed to improve the performance of estimating BP and to overcome the limitations of previous study. The method does not require a feature extraction process, which is a costly and laborious task, as ECG, PPG, and BCG signals were used as inputs after a simple bandpass filter was applied. In addition, the method enables a noninvasive and continuous BP monitoring as the physiological signals could be measured non-invasively. As mentioned in Section 3, the model with the attention mechanism outperformed others. These results imply that other physiological signal processing methods can be improved with the attention mechanism. In the future study, we will analyze which patterns of signals were weighted by the attention mechanism. In addition, we will consider a sufficiently large number of subjects and long-term data to optimize the model and improve performance.

## Figures and Tables

**Figure 1 sensors-20-02338-f001:**
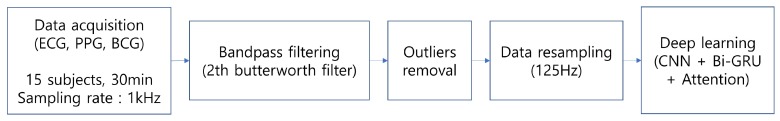
Brief flow chart of our experiment.

**Figure 2 sensors-20-02338-f002:**
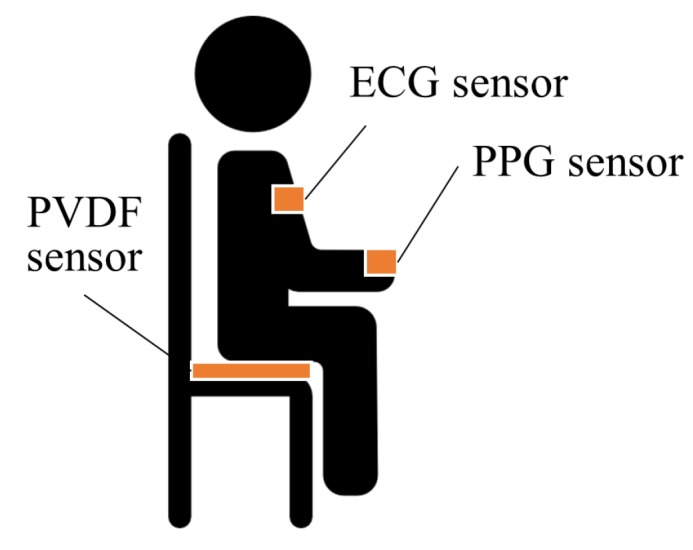
Overview of our experimental setup.

**Figure 3 sensors-20-02338-f003:**
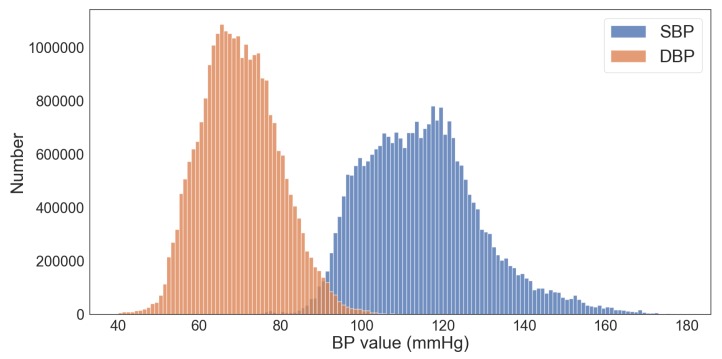
Distribution of BP values in the dataset used in this study.

**Figure 4 sensors-20-02338-f004:**
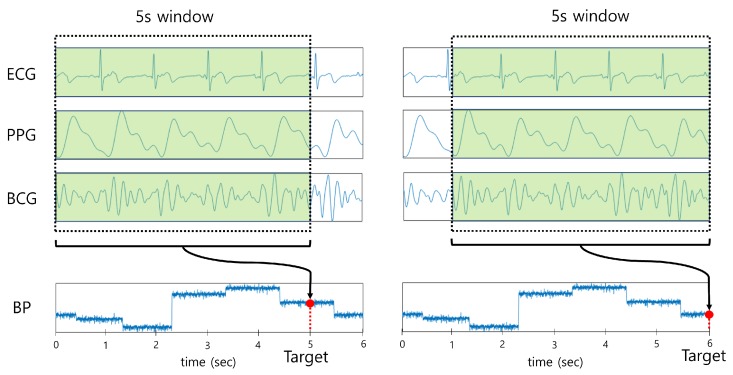
Data preprocessing of the deep learning model.

**Figure 5 sensors-20-02338-f005:**
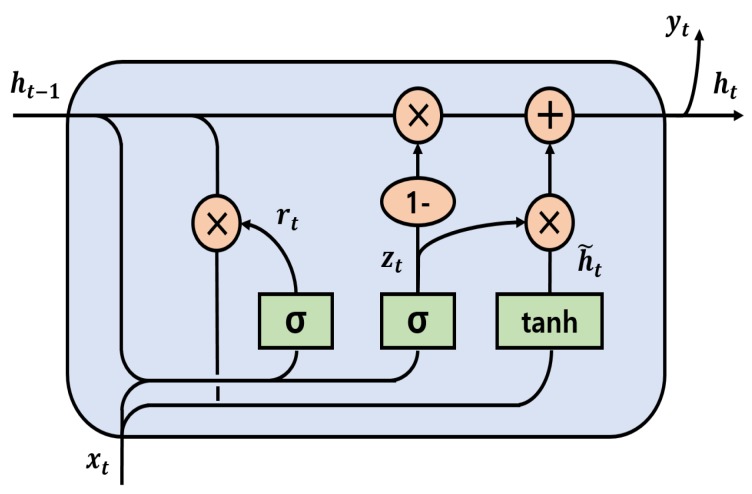
Internal structure of GRU.

**Figure 6 sensors-20-02338-f006:**
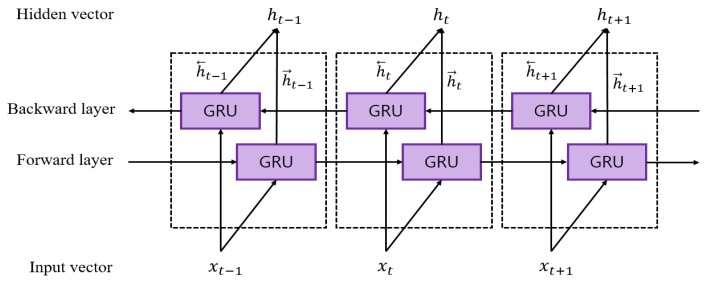
Bidirectional GRU structure.

**Figure 7 sensors-20-02338-f007:**
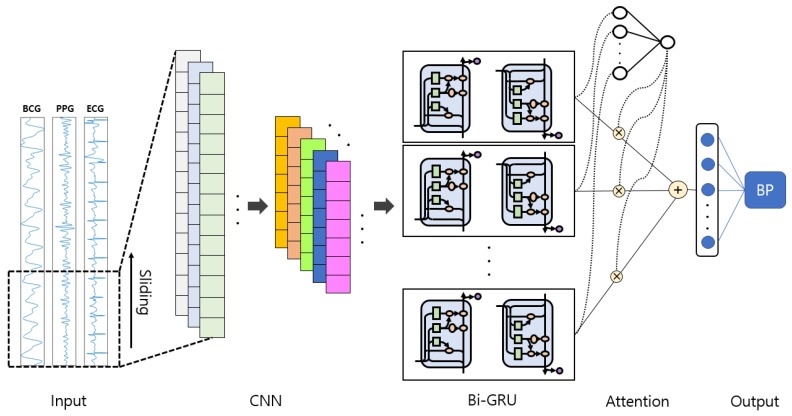
Overall structure of the proposed network.

**Figure 8 sensors-20-02338-f008:**
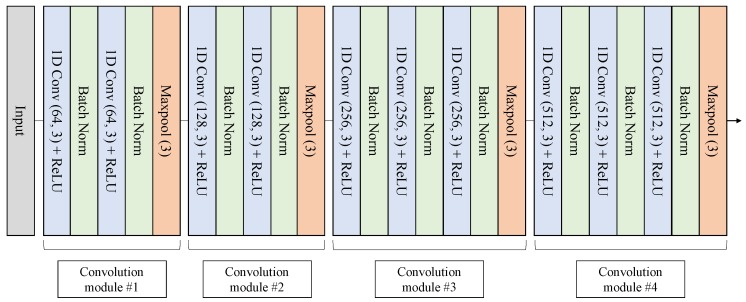
CNN structure of the proposed model.

**Figure 9 sensors-20-02338-f009:**
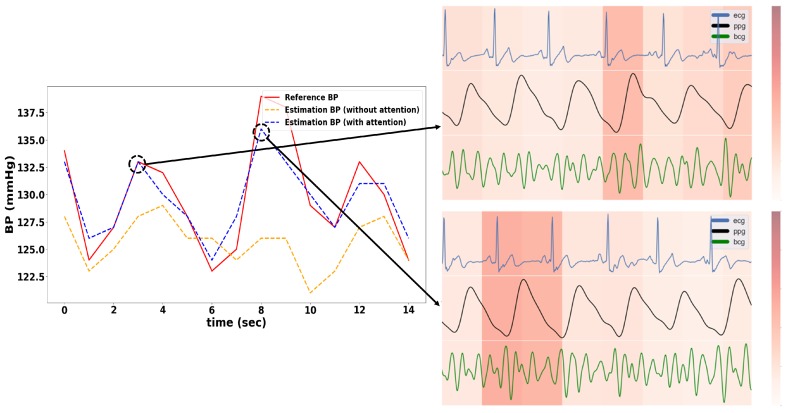
Left: Sample of the estimated BP with and without attention mechanism; Right: Heat map of the weights of the attention mechanism at the point where the error was low. The darker color denotes higher attention weight.

**Figure 10 sensors-20-02338-f010:**
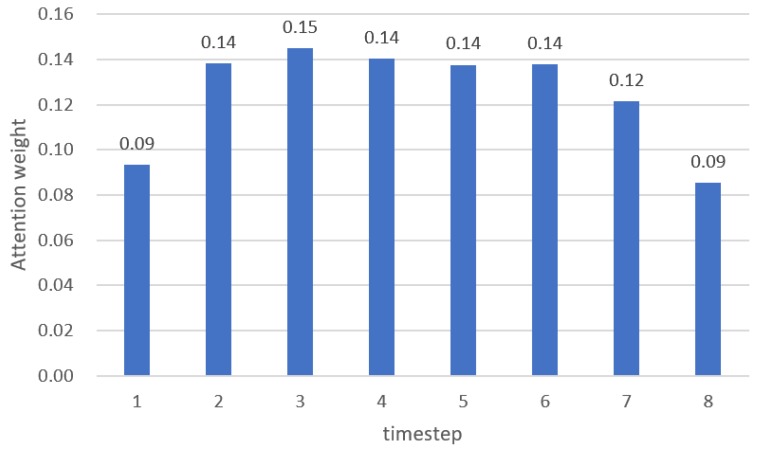
Mean attention weight across all datasets for each timestep.

**Figure 11 sensors-20-02338-f011:**
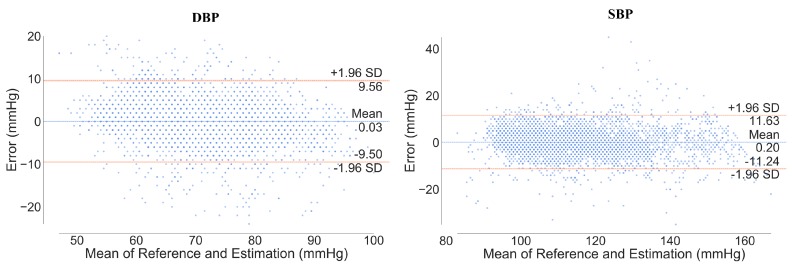
Bland–Altman plot of DBP and SBP. The orange line denotes the limit of agreement (LOA) and the blue line denotes the mean of difference error between reference and estimation.

**Figure 12 sensors-20-02338-f012:**
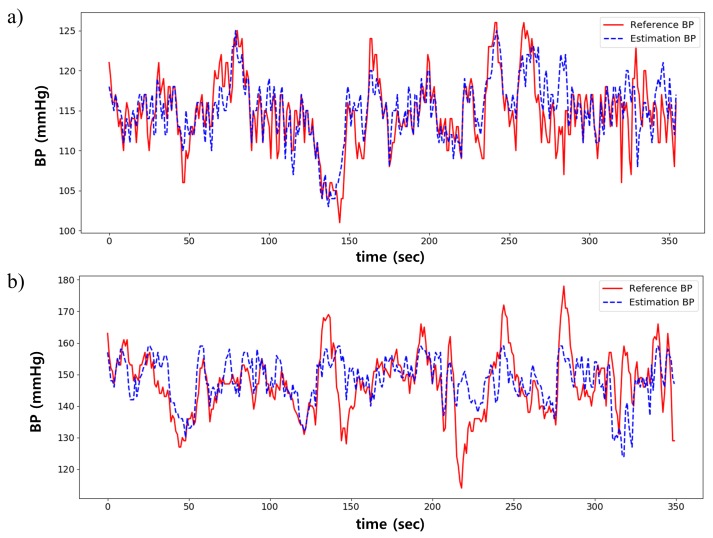
Comparison between estimated and reference BP. (**a**) is the best case; (**b**) is the worst case.

**Figure 13 sensors-20-02338-f013:**
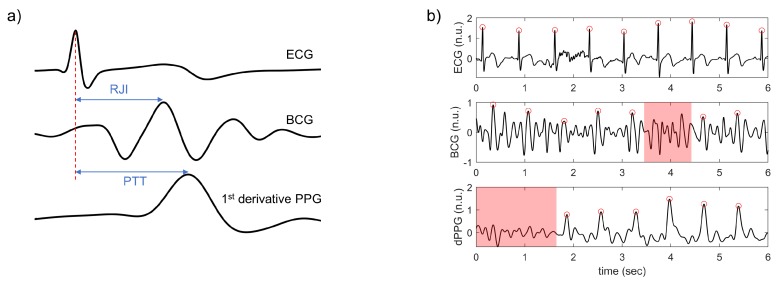
(**a**) example of the calculation of PTT and RJI in one cardiac cycle; (**b**) example of the excluded peaks. Red dots denote each characteristic point, and the red shaded region shows the area where peaks were not detected.

**Figure 14 sensors-20-02338-f014:**
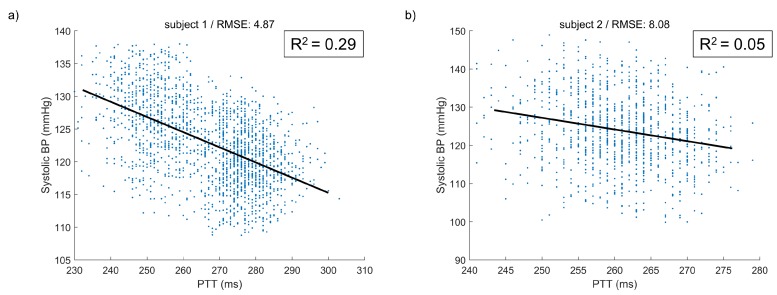
Scatter plots between PTT and Systolic BP of (**a**) good case and (**b**) bad case. The black line indicates a fitting line.

**Table 1 sensors-20-02338-t001:** The cutoff frequency of the filter applied to each signal.

Signal	HPF (Hz)	LPF (Hz)
ECG	0.5	35
BCG	4	15
PPG	0.5	15

**Table 2 sensors-20-02338-t002:** Detailed structure of the proposed model.

Network	Layer	Shape	Out	Padding	Stride	Kernel
CNN	Conv	625×3	64	Same	1	3
	BN + ReLU					
	Conv	625×64	64	Same	1	3
	BN + ReLU					
	Maxpool (size = 3)	625×64	-	Same	3	-
	Conv	209×64	128	Same	1	3
	BN + ReLU					
	Conv	209×128	128	Same	1	3
	BN + ReLU					
	Maxpool (size = 3)	209×128	-	Same	3	-
	Conv	70×128	256	Same	1	3
	BN + ReLU					
	Conv	70×256	256	Same	1	3
	BN + ReLU					
	Conv	70×256	256	Same	1	3
	BN + ReLU					
	Maxpool (size = 3)	70×256	-	Same	3	-
	Conv	24×256	512	Same	1	3
	BN + ReLU					
	Conv	24×512	512	Same	1	3
	BN + ReLU					
	Conv	24×512	512	Same	1	3
	BN + ReLU					
	Maxpool (size = 3)	24×512	-	Same	3	-
Bi-GRU	Forward	8×512	64		-	
	Backward	8×512	64		-	
	Concatenation					
Attention	1-layer perceptron	8×128	1		-	
	Activation tanh					
	Softmax					
	Weighted sum					
	1-layer perceptron	128	2		-	

**Table 3 sensors-20-02338-t003:** Performance comparison for combinations of input signals without attention model and with attention. The 95% confidence interval is indicated below the error of the proposed model.

Model	Input	SBP (mmHg)				DBP (mmHg)			
		**RMSE**	**MAE**	**SD**	$R^{2}$	**RMSE**	**MAE**	**SD**	$R^{2}$
CNN+Bi-GRU	ECG	7.02	5.51	4.66	0.24	5.16	4.06	3.45	0.27
	PPG	6.88	5.34	4.60	0.28	5.73	4.45	4.09	0.14
	BCG	7.24	5.59	5.03	0.20	5.29	4.06	3.71	0.22
	ECG, PPG	5.83	4.46	4.06	0.46	4.74	3.70	3.37	0.38
	ECG, BCG	6.74	5.30	4.60	0.31	4.82	3.74	3.27	0.34
	PPG, BCG	6.44	4.86	4.50	0.36	5.04	3.88	3.62	0.27
	ECG, PPG, BCG	5.87	4.51	4.14	0.48	4.73	3.71	3.39	0.40
**CNN+Bi-GRU** **+Attention** **(proposed model)**	**ECG, PPG, BCG**	**5.42** **[1.97, 8.87]**	**4.06** **[1.53, 6.59]**	**4.04**	**0.52**	**4.30** **[0.94, 7.72]**	**3.33** **[0.61, 6.05]**	**3.42**	**0.49**

**Table 4 sensors-20-02338-t004:** Mean values of RMSE, MAE, and $R^{2}$ when the input was a single signal and when it was a combination of multiple signals.

Input	SBP (mmHg)			DBP (mmHg)		
	**RMSE**	**MAE**	**mean** $R^{2}$	**RMSE**	**MAE**	**mean** $R^{2}$
Single signal	7.04	5.47	0.24	5.39	4.19	0.21
Multiple signals	6.21	4.78	0.40	4.83	3.76	0.35

**Table 5 sensors-20-02338-t005:** Results of paired *t*-test between various inputs for SBP estimation.

Inputs	ECG	PPG	BCG	ECG, PPG	ECG, BCG	BCG, PPG	ECG, BCG, PPG	Proposed Model
**ECG**		-	-	*p* < 0.05	-	-	*p* < 0.05	*p* < 0.05
**PPG**			-	*p* < 0.05	-	*p* < 0.05	*p* < 0.05	*p* < 0.05
**BCG**				*p* < 0.05	-	*p* < 0.05	*p* < 0.05	*p* < 0.05
**ECG, PPG**					*p* < 0.05	-	-	*p* < 0.05
**ECG, BCG**						-	*p* < 0.05	*p* < 0.05
**BCG, PPG**							-	*p* < 0.05
**ECG, BCG, PPG**								*p* < 0.05

**Table 6 sensors-20-02338-t006:** Results of paired *t*-test between various inputs for DBP estimation.

Inputs	ECG	PPG	BCG	ECG, PPG	ECG, BCG	BCG, PPG	ECG, BCG, PPG	Proposed Model
**ECG**		-	-	-	*p* < 0.05	-	-	*p* < 0.05
**PPG**			-	*p* < 0.05	*p* < 0.05	*p* < 0.05	*p* < 0.05	*p* < 0.05
**BCG**				-	-	-	-	*p* < 0.05
**ECG, PPG**					-	-	-	*p* < 0.05
**ECG, BCG**						-	-	*p* < 0.05
**BCG, PPG**							-	*p* < 0.05
**ECG, BCG, PPG**								*p* < 0.05

**Table 7 sensors-20-02338-t007:** Comparison between proposed model and MLR model.

Model	SBP (mmHg)				DBP (mmHg)			
	RMSE	MAE	SD	$R^{2}$	RMSE	MAE	SD	$R^{2}$
**Proposed model**	5.42	4.06	4.04	0.52	4.30	3.33	3.42	0.49
**MLR**	6.40	5.19	3.45	0.26	4.75	3.85	2.69	0.22

**Table 8 sensors-20-02338-t008:** Performance comparison with the AAMI standard.

		Mean Error	Standard Deviation
AAMI standard	SBP, DBP	**≤ 5 (mmHg)**	**≤ 8 (mmHg)**
**Proposed model**	**SBP**	**−0.20**	**5.83**
	**DBP**	**−0.02**	**4.91**

**Table 9 sensors-20-02338-t009:** Performance comparison with the BHS standard.

		Absolute Difference			Grade
		≤ 5 (mmHg)	≤ 10 (mmHg)	≤ 15 (mmHg)	
BHS standard	SBP, DBP	60%	85%	95%	A
		50%	75%	90%	B
		40%	65%	80%	C
		Worse than C			D
**Proposed model**	**SBP**	**73%**	**93%**	**98%**	**A**
	**DBP**	**80%**	**96%**	**99%**	**A**

**Table 10 sensors-20-02338-t010:** Performance comparison with related works.

Author	Data Size	Calibration	Model	Input		SBP (mmHg)	DBP (mmHg)
				Inputs	Signal	Error	Error
Chan et al. [14]	Unspecified	Cal-based	Linear regression	Feature (PTT)	ECG PPG	ME: 7.49 STD: 8.82	ME: 4.08 STD: 5.62
Kachuee et al. [15]	1000 subjects 10 min (MIMIC 3)	Cal-based	AdaBoost	Features	ECG PPG	MAE: 8.21 STD: 5.45	MAE: 4.31 STD: 3.52
		Cal-free				MAE: 11.17 STD: 10.09	MAE: 5.35 STD: 6.14
Kurylyak et al. [17]	15,000 heartbeats	Cal-based	Deep learning (ANN)	Features	PPG	ME: 3.80 STD: 3.46	ME: 2.21 STD: 2.09
Lee et al. [13]	30 subjects	Cal-based	Deep learning (ANN)	Feature (IPD)	BCG	ME: 0.01 STD: 6.75	ME: 0.05 STD: 5.83
Slapnivcar et al. [19]	510 subjects 700 h (MIMIC 3)	Cal-based	Deep learning (ResNet)	Raw	PPG	MAE: 9.43	MAE: 6.88
		Cal-free				MAE: 15.41	MAE: 12.38
Su et al. [16]	84 subjects 10 min	Cal-based	Deep learning (RNN)	Features	ECG PPG	RMSE: 3.73	RMSE: 2.43
Tanveer et al. [20]	39 subjects (MIMIC 1)	Cal-based	Deep learning (ANN+ LSTM)	Raw	ECG PPG	RMSE: 1.27 MAE: 0.93	RMSE: 0.73 MAE: 0.52
Wang et al. [18]	58,795 intervals of PPG (MIMIC 1)	Cal-based	Deep learning (ANN)	Features	PPG	MAE: 4.02 STD: 2.79	MAE: 2.27 STD: 1.82
This study	15 subjects 30 min	Cal-based	Deep learning (CNN+ Bi-GRU)	Raw	BCG	ME: −0.82 STD: 7.50	ME: −0.97 STD: 5.36
					ECG PPG	MAE: 4.46 STD: 4.06	MAE: 3.70 STD: 3.37
			Deep learning (CNN+ Bi-GRU+ Attention)		ECG PPG BCG	MAE: 4.06 STD: 4.04	MAE: 3.33 STD: 3.42

**Table 11 sensors-20-02338-t011:** Performance comparison between calibration-free and calibration-based methods using the proposed model.

Input	Method	SBP (mmHg)				DBP (mmHg)			
		**RMSE**	**MAE**	**SD**	$R^{2}$	**RMSE**	**MAE**	**SD**	$R^{2}$
ECG, PPG, BCG	Cal-based	5.42	4.06	4.04	0.52	4.3	3.33	3.42	0.49
	Cal-free	13.14	9.70	8.86	0.23	7.55	5.79	4.84	0.44

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
