# Peer review of "End-To-End Deep Learning Architecture for Continuous Blood Pressure Estimation Using Attention Mechanism"

_sensors, 2020, doi:10.3390/s20082338_

Round 1
Reviewer 1 Report
The authors propose a deep-learning architecture method consisted of a convolutional neural network (CNN) layer, a bidirectional gated recurrent unit layer and an attention layer for continuous blood pressure monitoring of patients with hypertension.
The paper structure is lacking critical information, which makes it hard to follow:
- The last part of the introduction section which analyses the related work is rather limited and needs expansion.
- In lines 68-69, the authors write: “In this study, we measured all signals (ECG, PPG, and BCG) used in the previous study and developed an end-to-end deep learning model for blood pressure estimation.”. This sentence is very confusing because they haven’t explained in the introduction section what is the previous study.
- The full name of each abbreviation must be written in the paper when each abbreviation is first introduced. Omitting this, makes the document hard to follow. The abbreviation table at the end of the document is useful, but makes the text difficult to follow because the reader has to go at the end of the paper every time an abbreviation is introduced.
- The authors should include a detailed flowchart depicting the overall method, in the beginning of section 2, and then follow this flowchart in the rest section 2. Otherwise, it is very difficult to “position” each part in the over analysis.
- Following the previous comment:
- Section 2.2.1. (Multiple Linear Regression Model) describes the model used for comparison. Why it is included in the main method?
- Sections 2.3.1, 2.3.2 and 2.3.3 must be moved before the section explaining the proposed architecture.
- Section 2.4 is lacking critical information about the initialization and execution of each experiment (e.g. how the initial weights and thresholds where initialized). A table summarizing the parameters for the experiments must be added. The authors should also include the number of hidden layers, the size of hidden layers and activation function types of hidden neurons in the parameters table. The table followed by an explanation would give a better understanding of the parameters used.
- The authors should include a comparative study, comparing their research with similar studies presented in the literature.
- In section 4.2 the authors clearly state that “the number of subjects was not sufficient and most of the subjects were healthy 20-year olds with no diseases”. Although I do appreciate the honesty, this statement totally undermines the importance and impact of the study. The authors must either include more subjects in the study or explain why their research is important (in terms of sample size). Also, they should add a column in the comparative study matrix (from the previous comment) for “sample size”, and compare the sample size of other studies presented in the literature vs their own.
- The language needs polishing since it contains grammar and syntactic errors.
Based on the above, the manuscript cannot be accepted in its current form (reject, but resubmission after major revision is encouraged).
Author Response
Thank you for your useful comments. Please find our responses in the word file attached. The changes that have been made in the manuscript was highlighted.

Reviewer 2 Report
This paper proposed a CNN network with attention mechanism for non-invasive blood pressure estimation. I have the following concerns:
- I suggest that you add some more results. Some more theoretical Math analysis, some simulation results and some comparison of the presented scheme with other schemes. May be some figures for the simulation results or the comparisons.
- More motivation/context regarding the application side of it, particularly on the aspects that make this technique particularly suited for industrial application scenarios, and how it would be applied in real scenarios. These aspects could additionally be supported with some related work context. Such as discussion of related work: Continuous Cuff-Less Blood Pressure Estimation Based on Combined Information Using Deep Learning Approach
- Clinical impact of blood pressure should be discussed deeply as well. such as Analysis of beat-to-beat blood pressure variability response to the cold pressor test in the offspring of hypertensive and normotensive parents.
- Comparsion with recent study and methods would be appreciated
- The abstract can be rewritten to be more meaningful. The authors should add more details about their final results in the abstract. Abstract should clarify what is exactly proposed (the technical contribution) and how the proposed approach is validated.
Author Response

(The authors gave the same response as above.)

Reviewer 3 Report
Section 2.1: ECG recording details must be provided (e.g., was DRL used ? electrode position?)
Was there any synchronization between ECG, PCG, and continuous BP monitoring?
Section 2.2: Were there any outliers in the data? How were they identified and treated?
2.2.1: Which R-peak detection algorithm (QRS detection) was used for ECG? The same for PCG and PCG.
Line 102: which percentage of the data was discarded.
Figure 3.a: please indicate that the y-axis unit is arbitrary.
Section 2.3: did the authors implement the code or use any available packages and with which programming language?
Section 2.4: How were the learning rate and decay parameters estimated?
Section 2.4: A critical issue:
It is necessary to train the system on some subjects and test it on others and report the results. If such results are not accurate, it must be mentioned as one of the limitations of the proposed method that is not “calibration-free.” Moreover, in the literature review, new calibration-free methods must be cited.
The other important issue in Section 2.4:
The authors used a hold-out validation (70%). This method is biased and introduces the Type-III statistical error. Either cross-validation or repeated-holdout must be used. Accordingly, MEAN and SD values must be reported in results Tables 2 and 3. Moreover, instead of CC, R-square values must be reported. Following the TRIPOD recommendation:
https://www.equator-network.org/reporting-guidelines/tripod-statement/
The CI 95% of the indices must be provided.
Proper statistical tests must be used to identify which method outperforms the others (proposed methods vs. other combinations vs. MLR).
In Fig.13, what are the R-square values?
Any running time analysis?
Moreover, since no hypertension and hypotension data were used in this study, the BP range must be mentioned in the abstract, showing the range of the data used in this study. Moreover, if the subject cross-validation does not provide accurate results, it must be directly mentioned in the abstract that the method is not calibration-free.
Three very positive points of the study:
The research question is fundamental and very practical.
The Bland Altman plot was used in the study.
The clinical standards were used to categorize the effectiveness of the method.
Author Response

(The authors gave the same response as above.)

Round 2
Reviewer 1 Report
The authors have addressed all my previous comments, significantly improving their work.
Minor comment: Replace "ours" in last row of Table 10 with "This study".
Author Response
Thank you for your useful comments.
Minor comment: Replace "ours" in last row of Table 10 with "This study".
Response: The word "Ours" in Table 10 was replaced to "This study".
Reviewer 2 Report
all my concerns have been addressed
Author Response
Thank you for your useful comments.
Reviewer 3 Report
The authors addressed the issues raised by the reviewer. However, regarding to the paired t test, as repeating a test increases type I error, repeated measurement ANOVA or GEE with proper post hoc must be used.
Author Response
Thank you for your useful comments.
Point: Regarding to the paired t test, as repeating a test increases type I error, repeated measurement ANOVA or GEE with proper post hoc must be used.
Response: We used repeated measurement ANOVA (RM-ANOVA) test at the first time, and the results showed that the differences between the models were significant (p<0.01). However, the result of a paired t-test was shown in the manuscript as we wanted to investigate that the difference between the proposed model and other model was significant.
The statement regarding the RM-ANOVA test was added in the manuscript.
(line 192) A repeated measurement analysis of variance (ANOVA) test was used to compare the performance when the combination of inputs was varied, and the differences between models were significant in both SBP and DBP estimation (p < 0.01). In addition, a paired t-test was performed between the results to compare each models.